# OpenReview forum: "MetaRuleReasoner: Beyond Chain-of-Thought—Neural Rule-Based Reasoning for Reliable Mathematical Computation"
_ICLR.cc/2026/Conference — ICLR 2026 Conference Withdrawn Submission_

### Official Review · Reviewer_X3zF · 2025-11-01

**Soundness:** 2
**Presentation:** 2
**Contribution:** 2
**Rating:** 4
**Confidence:** 4

**Summary:**

This paper proposes "neural rule-based reasoning" as an alternative to chain-of-thought (CoT) reasoning for mathematical computation in language models. The authors assert that their MetaRuleReasoner approach achieves perfect accuracy on arithmetic tasks where CoT methods exhibit performance degradation. The method purportedly integrates the reliability of symbolic AI with the flexibility of neural networks through systematic learning and application of explicit computational rules.

**Strengths:**

1. The paper successfully identifies genuine limitations of chain-of-thought approaches, especially concerning reliability in systematic domains such as arithmetic.

2. The conceptual differentiation between pattern-based and rule-based reasoning presents an interesting avenue worthy of investigation.

3. The manuscript demonstrates clear writing and solid structural organization.

**Weaknesses:**

1. Questionable technical originality: Although the paper frames its approach as fundamentally novel, the underlying concept of training neural networks to execute algorithmic procedures has established precedent. Comparable methodologies appear throughout the Neural Algorithmic Reasoning, Neural Program Induction, and Neuro-symbolic AI literature.

2. Constrained task environment: The evaluation concentrates on arithmetic, which while genuinely rule-based, constitutes an exceedingly narrow subset of mathematical reasoning. The paper provides no evidence of applicability to more sophisticated domains requiring higher-order reasoning capabilities.

**Questions:**

1. **ADD MORE COMPARISION PLEASE**. The paper positions itself as entirely separate from neuro-symbolic approaches while failing to adequately differentiate itself from recent developments in this domain.

2. **Missing ablation analysis**: The absence of component-wise performance analysis prevents understanding which elements contribute most significantly to results. Please add MORE ABLATION analysis

---

### Official Review · Reviewer_PttA · 2025-11-01

**Soundness:** 2
**Presentation:** 1
**Contribution:** 2
**Rating:** 2
**Confidence:** 4

**Summary:**

This paper introduces "neural rule-based reasoning" as an alternative to chain-of-thought reasoning for mathematical computation in large language models. MetaRuleReasoner achieves 100% accuracy on multi-digit arithmetic tasks by applying learned computational rules systematically, while CoT models (including GPT-4) show performance degradation with increasing complexity. Unlike CoT's pattern-based approach that generates natural language explanations, neural rule-based reasoning decomposes problems into explicit rule applications with systematic composition and verification at each step.

**Strengths:**

- The paper is clearly written.

**Weaknesses:**

- The paper format violates ICLR'26 requirements.
- Poor baselines. Experiments are conducted with non-reasoning LLMs (such as GPT-4, Google-PaLM, Qwen-72B). How does MetaRuleReasoner's performance compare to OpenAI's O-series models or Google Gemini Thinking?

**Questions:**

Please refer to the "Weaknesses" Section

---

### Official Review · Reviewer_sKsU · 2025-11-01

**Soundness:** 2
**Presentation:** 2
**Contribution:** 1
**Rating:** 2
**Confidence:** 5

**Summary:**

This paper introduces MetaRuleReasoner, a neural rule-based reasoning approach that aims to address limitations of chain-of-thought (CoT) reasoning in mathematical computation. While CoT generates natural language reasoning steps probabilistically, the proposed approach applies learned computational rules systematically with explicit rule application and verification at each step. Experiments demonstrate that MetaRuleReasoner achieves 100% accuracy on multi-digit arithmetic tasks, while CoT models like GPT-4 show performance degradation as problem complexity increases.

**Strengths:**

- The paper contains good motivations.
- The paper is easy-to-follow.

**Weaknesses:**

- Formalization of this paper seems to violate ICLR's rules.
- Baselines of the paper seem out-of-date. Authors should consider adding more strong baselines.

**Questions:**

- Why can MetaRuleReasoner achieve 100% accuracy across all tasks? Is this overfitting?
- Can MetaRuleReasoner generalize to other math-related tasks?

---

### Official Review · Reviewer_7569 · 2025-11-04

**Soundness:** 1
**Presentation:** 1
**Contribution:** 1
**Rating:** 0
**Confidence:** 5

**Summary:**

This paper introduces MetaRuleReasoner, a neural rule-based reasoning system as a distinct alternative that achieves systematic reliability through explicit rule application and complete domain coverage.

**Strengths:**

maybe nothing

**Weaknesses:**

1. None of the figures or tables include hyperlinks.
2. The paper is not using the ICLR template. I’m unsure whether this alone warrants a desk reject, but it could introduce potential unfairness in overall paper length.
3. The motivation in the introduction (the three deficiency claims) has no supporting citations or experimental evidence.
4. The writing feels rushed, the references are very sparse, and most are from 2023–2024; given the rapid progress in CoT techniques, the paper’s timeliness is questionable. I don’t think this is an appropriate submission for ICLR 2026.
5. The method’s generality is clearly very narrow; the paper focuses on toy tasks, and the models are quite outdated—for example, it uses Llama 2 and Qwen—while the latest reasoning models are not considered at all.

**Questions:**

N/A

**Details Of Ethics Concerns:**

Template Usage

---

### Note · Authors · 2025-12-15

I have read and agree with the venue's withdrawal policy on behalf of myself and my co-authors.